# Extracellular Vesicles: A New Frontier for Cardiac Repair

**DOI:** 10.3390/pharmaceutics14091848

**Published:** 2022-09-01

**Authors:** Benshuai You, Yang Yang, Zixuan Zhou, Yongmin Yan, Leilei Zhang, Jianhua Jin, Hui Qian

**Affiliations:** 1Wujin Institute of Molecular Diagnostics and Precision Cancer Medicine of Jiangsu University, Wujin Hospital Affiliated with Jiangsu University, Changzhou 213017, China; 2Jiangsu Key Laboratory of Medical Science and Laboratory Medicine, School of Medicine, Jiangsu University, Zhenjiang 212013, China; 3Clinical Laboratory Center, The Affiliated Taizhou People’s Hospital of Nanjing Medical University, Taizhou 225317, China

**Keywords:** extracellular vesicles, heart disease, mesenchymal stem cells, targeted delivery

## Abstract

The ability of extracellular vesicles (EVs) to regulate a broad range of cellular processes has recently been used to treat diseases. Growing evidence indicates that EVs play a cardioprotective role in heart disease by activating beneficial signaling pathways. Multiple functional components of EVs and intracellular molecular mechanisms are involved in the process. To overcome the shortcomings of native EVs such as their heterogeneity and limited tropism, a series of engineering approaches has been developed to improve the therapeutic efficiency of EVs. In this review, we present an overview of the research and future directions for EVs-based cardiac therapies with an emphasis on EVs-mediated delivery of therapeutic agents. The advantages and limitations of various modification strategies are discussed, and possible opportunities for improvement are proposed. An in-depth understanding of the endogenous properties of EVs and EVs engineering strategies could lead to a promising cell-free therapy for cardiac repair.

## 1. Introduction

Heart disease is a major cause of death worldwide, especially among aging populations. Approximately 17.8 million deaths globally in 2017 were caused by heart disease [1]. Thus, effective therapy for cardiac repair is required. Extracellular vesicles (EVs), nanoscale small vesicles, are secreted by many types of cells and play a key role in intercellular communication [2,3]. “EVs” is a collective term for various subtypes of membrane components released by cells, including exosomes (30–150 nm), microvesicles (200–2000 nm), and apoptotic bodies (1–10 μm) [4]. EVs contain abundant cargo such as proteins, nucleic acids, and lipids [5,6,7]. Compared with parental cells, EVs retain the function of the host and also have the advantages of low immunogenicity, long-term stability, easy storage, and biocompatibility [8,9]. EVs can easily diffuse to the blood, penetrate tissues through the circulatory system, and even cross the blood–brain barrier [10]. Therefore, developing novel strategies to optimize such cell-free nanoparticles could profoundly affect clinical practice.

Studies have revealed the benefits of mesenchymal stem cells (MSCs) and other cells in the treatment of cardiac disease, with the effects of paracrine playing an essential role in these benefits [11,12]. For example, conditioned medium from adipose-derived stem cells was able to attenuate ischemia-reperfusion-induced apoptosis and fibrosis through the miR-221/222/ PUMA/ETS-1 pathway in a mouse model [13]. As the main paracrine component of cells, EVs exhibit regenerative properties in myocardial infarction, ischemia-reperfusion injury, and viral myocarditis [14]. Because of their acellular nature, EVs even demonstrate greater myocardial reparative capabilities than parent cells and are also safer [15]. A large body of evidence suggests that many bioactive molecules in EVs play a part in the repair processes by promoting angiogenesis, modulating immune responses and inhibiting apoptosis. The use of engineering methods can further amplify the therapeutic effects of EVs by increasing their targeting, bioactivity, retention, and yield.

In this review, we evaluate the feasibility of using natural or engineered EVs for treating myocardial injury. We present the therapeutic effects of EVs from different sources when applied to various myocardial injury models and focus on the role of multiple bioactive molecules within EVs in mediating cardioprotection. We discuss different engineering strategies and highlight the crucial challenges and opportunities in clinical translation.

## 2. Biosynthesis and Characteristic of EVs

Because the specific indicators of EV subtypes are still being debated, it is recommended to define EVs using operational terms on the basis of physical characteristics (e.g., size and density) or the biochemical composition or descriptions of conditions or cell of origin. For example, EVs can be divided into small EVs (sEVs, <100 nm or <200 nm) and medium/large EVs (m/lEVs, >200 nm). In previous studies, EVs have often been classified into three types: exosomes, microvesicles, and apoptotic bodies [16]. The term “exosomes” has been used frequently in previous studies, but strictly speaking, it should only be used when data demonstrating endosomal origin of an EV is provided. In this review, in order to be consistent with the original article, the names of EVs that appeared in the original article were used. Exosomes are primarily released by the fusion of multivesicular bodies (MVBs) and plasma membrane, and microvesicles and apoptotic bodies are generated through the direct budding of cell membrane [17]. The specific formation process of exosomes and microvesicles will be described below. The host cell-derived cargo carried by EVs is responsible for their diverse biological functions. Currently, research on the regeneration function of EVs mainly focuses on exosomes and microvesicles, with little attention given to apoptotic bodies. EVs deliver their contents to recipient cells through endocytosis, ligand–receptor interaction, and direct fusion with the plasma membrane (Figure 1). In recent years, some new types of EVs have been discovered, such as retractosomes [18]. However, little is known about their biogenesis and physiological function.

Several stages—including endocytosis, endosome and MVBs formation, and exosomes release—are involved in the formation of exosomes [19]. Although the exact process remains unclear, the biogenesis of exosomes has been investigated. First, the lipid raft domain of the plasma membrane forms early endosomes through the endocytosis pathway. With the assistance of the Golgi complex, these early endosomes mature into late endosomes, the bilayer membrane of which produces vesicles that accumulate in the lumen [20]. Both endosomal sorting complex required for transport (ESCRT)-dependent and ESCRT-independent pathways are involved in the process of MVBs formation [21,22]. MVBs then fuse with the plasma membrane to release exosomes or fuse with lysosomes and then degrade [23]. Once MVBs have fused with the plasma membrane, intraluminal vesicles are released into the extracellular environment.

Microvesicles are released into the extracellular space through outward sprouting of the plasma membrane [24]. The asymmetric distribution of phospholipids in the plasma membrane bilayer is involved in the formation of microvesicles. The outer membrane is dominated by phosphatidylcholine and sphingomyelin, whereas the intima is dominated by phosphatidylserine and phosphatidylethanolamine. Increased cytosolic calcium disrupts the phospholipid asymmetry by activating the phospholipid crawling enzyme and thus initiating cell budding. Ca2+-dependent proteolysis participates in the release of microvesicles through cytoskeleton degradation [25].

Given the small size of EVs and the complexity of their content, characterizing EVs accurately can be difficult and require multiple measurements. Currently, EVs are identified on the basis of their morphology, particle size, content, and surface markers [26]. EVs may collapse during fixation and dehydration, and this collapse gives them an artificial cup-shaped appearance, which can be observed using a transmission electron microscope [27]. Atomic force microscopy and cryoelectron microscopy are used to evaluate the substructural organization of EVs with high resolution [28,29]. Nanoparticle tracking analysis (NTA) is widely used to measure EVs and can provide information on their size distribution and particle concentration [30]. New detection technologies such as nanoflow cytometry are often applied in comprehensive analyses [31,32,33]. The development and application of specific biosensors provide a new approach for the bioanalysis of EVs [34,35]. Western blotting, mass spectrometry, and liquid chromatography are commonly adopted for protein assays [36,37,38]. New platform based on phage display are developed for characterization of EVs based on their different antigenic reactivities [39]. Polymerase chain reaction (PCR), high-throughput sequencing, and the newly developed droplet digital PCR are often employed for nucleic acid detection [40,41]. Moreover, chromatographic-mass spectrometry is the main method used for the lipidomic analysis of EVs [42]. A series of EVs surface proteins has been recognized as specific surface markers that can be used for EVs identification and are divided into transmembrane proteins (CD9, CD63, and CD81), heat shock proteins (Hsp70 and Hsp90), membrane fusion proteins (Alix, Annexin, and TSG101), membrane transporters (GTPases), and lipid-bound proteins [43,44]. It is also recommended to detect at least one negative protein marker such as calnexin, albumin in EVs according to the International Society for Extracellular Vesicles (ISEV) guidelines [45].

## 3. Natural EVs for Myocardial Injury Therapy

In the treatment of myocardial injury, stem cells are the most common donor cells selected as the source of EVs and include MSCs, embryonic stem cells (ESCs), induced pluripotent stem cells (iPSCs) and trophoblast stem cells (TSCs). Several studies have indicated that the repair that MSCs perform in myocardial injury is mainly mediated by the release of EVs [12]. MSCs-EVs were demonstrated to have a similar mechanism to MSCs in the repair of myocardial infarction. Shao et al. found that MSCs-derived exosomes had a stronger therapeutic effect than MSCs in a myocardial infarction model because of the significantly different expression of miRNAs from exosomes versus MSCs [46]. Current studies have shown that MSC-EVs aid the recovery of cardiac function mainly by inhibiting the death of cardiomyocytes, including apoptosis and ferroptosis. The EVs-mediated delivery of miRNAs (e.g., miR-23a-3p) and proteins (e.g., BMP2 and PINK1) plays a crucial role in the process [47,48,49]. In addition to directly inhibiting cardiomyocyte apoptosis, MSCs-EVs exert cardioprotective effects through the immunomodulation of macrophages. Zhao et al. demonstrated that MSCs-derived exosomes effectively reduced infarct size and alleviated inflammation in the heart after myocardial ischemia–reperfusion [50]. Further studies showed that exosomes-mediated miR-182 delivery to target TLR4 is essential in modifying the polarization of macrophages M1 to M2. Moreover, the miRNAs in MSCs-EVs such as miR-100-5p, miR-143-3p, miR-199a-3p, and miR-29a-3p also play functional roles in promoting the recovery of damaged myocardial tissue [51,52,53,54]. Other non-coding RNAs-such as long noncoding RNAs (lncRNA UCA1, lncRNA HCP5) and circular RNAs (circRNAs)-are involved in cardioprotective effects through the delivery of EVs [55,56,57].

Other stem-cell-derived EVs have also been utilized to treat heart disease. EVs derived from human-induced pluripotent stem-cell-derived cardiovascular progenitors were discovered to outperform cell injections and improve cardiac function in a myocardial infarction mouse model; this effect may have been associated with the abundant miRNAs in the tissue-repair pathways of EVs [58]. Ni et al. demonstrated that TSCs derived- exosomes were able to attenuate doxorubicin (Dox)-induced cardiac injury through anti-apoptotic and anti-inflammatory effects [59]. Further studies showed that this effect may be associated with an increase in ZEB1 expression and the inhibition of miR-200b expression in cardiomyocytes. Moreover, the research team found TSCs-derived exosomes mediated the delivery of let-7i to cardiac tissue, which inhibited myocardial apoptosis and mitigated cardiac fibrosis by downregulating YAP signaling [60]. Another study demonstrated the salutary effects of ESCs-derived exosomes on cardiac neovascularization and the reduction of apoptosis and fibrosis after myocardial infarction. ESCs-derived exosomes mediated the delivery of miR-294 to cardiac progenitor cells, thus promoting survival, cell cycle progression, and proliferation [61].

Other cells-derived EVs are also used for treating cardiovascular diseases. Under vasculogenic conditioning, subsets of proinflammatory cells in peripheral blood mononuclear cells (PBMCs) are beneficially transformed into proregenerative cells, collectively referred to as regeneration-associated cells (RAC) [62,63]. The systemic injection of RAC- EVs was superior to that of MSC-EVs for enhancing cardiac function because it delivered essential angiogenesis, antifibrosis, and anti-inflammatory miRNAs to myocardial ischemic tissue [64]. Dox and trastuzumab (Trz) are effective cancer drug treatments for HER2-positive breast cancer; the intravenous injection of cardiac-resident-mesenchymal-progenitor-cells (CPCs)-derived exosomes could attenuate Dox/Trz-induced cardiotoxicity, which includes the prevention of reactive oxygen species (ROS) production, reduction of inflammatory cell infiltration, and fibrosis [65]. Mechanism exploration revealed that miR-146a-5p that is enriched in exosomes may suppress target genes TRAF6 and MPO. In addition, CPCs-derived EVs could inhibit apoptosis in cardiomyocytes and enhance tube formation in endothelial cells by delivering miR-210 and miR-132 [66]. Many studies on EVs have been conducted in rodents; data on myocardial benefits in large animals are rare. Gallet et al. demonstrated that intramyocardial injection of exosomes secreted by cardiosphere-derived cells (CDCs) could effectively reduce scar formation, prevent adverse remodeling, and improve the left ventricular ejection fraction in acute and convalescent myocardial infarction pig models [67]. Gao et al. also demonstrated that human induced pluripotent stem-cells-derived exosomes showed benefit effects in recovery from pig myocardial infarction through improving left ventricular ejection fraction, cardiac hypertrophy, cell apoptosis, and angiogenesis [68].

The aforementioned studies focused on the role of exogenous EVs. The abundance of EVs in plasma motivated us to evaluate whether EVs play roles in mediating cardiac repair effects [69]. Studies have reported that a reduction in the infarction area through remote ischemic conditioning can be mediated by circulating EVs that accumulate in the damaged myocardium [70]. Consequently, researchers have explored the beneficial role of plasma-derived EVs in heart repair. Exosomes derived from the blood of rats and human volunteers were applied to a model of cardiac ischemia–reperfusion injury, and powerfully cardioprotective effects that were accompanied by activation of the HSP70–TLR4 communication axis were discovered [71]. To explore the intrinsic components of plasma circulating EVs, Wang and colleagues identified that miR-486 mediated the inhibition of PTEN, which then activated AKT and protected cardiomyocytes from apoptosis [72]. Additionally, the scholars used cardiac homing peptide (CHP) to link EVs and found that the infarct size in the heart was further diminished and the cardiac retention of EVs was increased in mice and dogs. Furthermore, Jin et al. observed that cotransplantation of bone marrow MSCs with plasma-derived EVs improved myocardial remodeling and promoted an increase in the myocardial capillary density by activating AKT signaling in a rat model of myocardial infarction [73].

To improve the effects of treatments, researchers have developed various methods for EVs administration. For overcoming the low efficiency of EV delivery to the heart, Yao et al. designed a minimally invasive exosome spray by using biomaterials and tested the spray’s feasibility and safety in mouse and pig models of acute myocardial infarction [74]. Compared with the injected exosomes, the exosomes spray exhibited high retention in the heart, improved cardiac function, reduced fibrosis, and promoted cardiac endogenous angiomyogenesis. In addition, it is also an interesting strategy of increasing the retention of EVs in the heart to prolong the therapeutic effects. Researchers have developed a hydrogel patch by encapsulating iPSC-derived cardiomyocytes-EVs into a 7 mm diameter collagen gel-foam mesh. The hydrogel patch was found to continually release EVs into the injured heart area, significantly promote recovery of the ejection fraction, decrease the burden of arrhythmia, and reduce the extent of cardiomyocyte apoptosis after infarction [75]. The development of these new methods further enhances the efficacy and feasibility of the clinical translation of EVs (Table 1).

## 4. Engineering EVs for Myocardial Injury Therapy

Although EVs are promising in the treatment of myocardial injury, many drawbacks exist in clinical translation such as heterogeneity and the limited tropism of EVs. Methods to increase the contents, targeting, bioactivity, kinetics, and biodistribution of EVs are necessary to overcome the shortcomings of natural EVs. Various methods have been developed to improve the treatment efficiency of EVs in heart disease. Modification of the donor cell or direct modification of the isolated EVs is often performed when bioengineering EVs (Figure 2).

### 4.1. Modification of the Donor Cell

Genetic engineering using viral or plasmid vectors is a convenient means of modifying parent cells and producing engineered EVs. Various therapeutic molecules, including miRNAs and proteins, can be encapsulated into secreted EVs by transfecting cells. For example, miR-21 has been reported to inhibit apoptosis by targeting the PDCD4/AP-1 pathway in infarcted cardiomyocytes [76,77]. In a mouse model of myocardial infarction, EVs derived from HEK293T cells overexpressing miR-21 effectively delivered miR-21 to cardiomyocytes and endothelial cells, thereby inhibiting their apoptosis and promoting improvement of cardiac function [78]. Compared with liposomes and polyethylenimine, EVs significantly inhibited the degradation of miR-21. In another study, exosomes were isolated from miRNA-181a-overexpressing umbilical cord blood-derived MSCs; compared with control exosomes, these exosomes exerted stronger repairing effects on myocardium ischemia-reperfusion injury through the inhibition of c-Fos [79]. Lin et al. showed that EVs secreted by β-catenin-engineered immortal CDCs inhibited the activation of NF-κB by delivering miR-4488, which consequently reduced the level of inflammation and suppressed arrhythmogenesis in arrhythmogenic cardiomyopathy [80]. Krüppel-like factors (KLFs) are highly expressed in endothelial cells and have anti-inflammatory effects [81]. Therefore, researchers have separated EVs from KLF2-overexpressing endothelial cells and determined whether the KLF2-EVs have a protective effect on cardiac injury. Excitingly, in both myocardial ischemia-reperfusion injury and Dox-induced dilated cardiomyopathy, KLF2-EVs exerted significant anti-inflammatory effects by targeting CCR2 to prevent Ly6Chigh monocytes recruitment to the myocardium [82,83]. Additionally, in myocardial infarction models, exosomes derived from Akt-, GATA4-, and TIMP2- modified MSCs demonstrated cardiac repair capabilities by promoting angiogenesis and inhibiting apoptosis [84,85,86].

A common strategy to increase the targeting ability of EVs is to express cardiac-targeting proteins on the surface of EVs through gene transfection. Lamp2b, a type of EVs surface protein, is widely used for displaying targeting motifs. Bone marrow MSCs transfected with a lentivirus of Lamp2b that was fused with ischemic myocardium-targeting peptide CSTSMLKAC (IMTP) produced exosomes that accumulated greatly at myocardial ischemia sites [87]. Additionally, the IMTP-exosomes significantly restored cardiac function by reducing inflammation and fibrosis and promoting angiogenesis. In another example, the researchers constructed a plasmid that expressed Lamp2b fused to a cardiomyocyte specific peptide (CMP), WLSEAGPVVTVRALRGTGSW and transfected it into CDCs [88]. Compared with control exosomes, the CDCs-derived exosomes expressed CMP on their surface and demonstrated higher cardiac retention.

EVs elicit different effects on recipient cells depending on the cultural environment of the parent cells. Changing the culture environment of donor cells is another effective method of harvesting therapeutic EVs. When processing a culture, oxygen tension is a key factor that affects the biological behavior of cells. Hypoxia induces the expression of cytoprotective genes in stem cells and improves their differentiation potential, which alters the content of EVs [89,90]. Studies have shown that hypoxia-preconditioned stem cell-derived EVs have significantly stronger myocardial repair effects than native EVs do. Wu et al. demonstrated that EVs secreted from human pluripotent stem-cells-derived cardiovascular progenitor cells under hypoxic conditions exhibited stronger cardioprotective effects than normal EVs in myocardial infarction [91]. The enhanced effects may be attributed to targeting miR-497 due to the over-enriched lncRNA MALAT1 in EVs under hypoxic conditions. Hypoxic conditions also increased the expression of lncRNA-UCA1 in MSCs-derived exosomes [92]. The hypoxic-conditioned exosomes exerted better cardioprotective effects against myocardial injury than did normal exosomes, and these effects were exerted through the miR-873-5p–XIAP axis. To further validate the effect of hypoxia preconditioning on EVs, researchers investigated the cardiac recovery effect of hypoxia-induced EVs in a nonhuman primate model. In a cynomolgus monkey myocardial infarction, EVs from hypoxia-preconditioned MSCs were enriched in miR-486-5p and promoted cardiac angiogenesis through fibroblast MMP19-VEGFA cleavage signaling [93]. Jung et al. investigated the molecular mechanism of EVs originating from human iPSC-derived cardiomyocytes in myocardial self-repair. They discovered that hypoxic conditions resulted in significantly higher abundance of miR-106a-363 in EVs and promoted a restart of the cardiomyocyte cell cycle by inhibiting the Notch3 pathway, which consequently improved heart function [94]. These studies highlight the positive effect of altering oxygen tension on the cardiac repair potential of EVs.

Exogenous stimuli are another novel strategy used to optimize EVs through coincubation with parental cells and to deliver bioactive molecules to injured cardiomyocytes. For example, Zhuang et al. obtained exosomes from macrophage migration inhibitory factor (MIF)-pretreated MSCs (exosomeMIF) and tested the cardioprotective effects on Dox-induced cardiomyopathy [95]. The results indicated that exosomeMIF restored cardiac function and exerted anti-senescence effects through LncRNA-NEAT1 transfer, partially by inhibiting miR-221-3p and activating SIRT2. Another study reported that by promoting endothelial cell function, atorvastatin-pretreated MSCs-derived exosomes were able to significantly improve the recovery of cardiac function in rats with acute myocardial infarction. Consequently, the infarct size and cardiomyocyte apoptosis were found to decrease, which may be attributed to overexpressed lncRNA H19 in the exosomes [96]. In one report, hemin-pretreated MSCs-derived exosomes were enriched in miR-183-5p and significantly improved cardiac function and reduced fibrosis compared with blank exosomes [97]. Mechanistically, the aforementioned exosomes attenuated mitochondrial fission and the cellular senescence of cardiomyocytes by regulating the HMGB1/ERK pathway in a mouse model of myocardial infarction. However, the exact mechanisms to determine the functional cargoes enriched in EVs after exogenous stimuli remain unclear. It limits the abroad application for further discovery of unknown stimuli.

### 4.2. Direct Modification of Isolated EVs

Manipulating parental cells to produce engineered EVs can result in most of the biophysical properties of EVs being retained. However, parental cell manipulation may cause unforeseen consequences for the biology of EVs and ultimately interfere with EVs biogenesis and alter other biological properties. Direct modification of isolated EVs does not change the biological origin of EVs, and various methods can be used to directly impart the desired biological properties as necessary.

Many studies have shown that miRNA in EVs has a therapeutic effect by inhibiting pathological genes after myocardial injury. Therefore, electroporation of miRNA into EVs may be a promising engineering strategy to increase the therapeutic efficacy of EVs. In one study, cardiac-progenitor-cells-derived exosomes were transfected with proangiogenic miR-322 by using electroporation. The cardiac therapeutic efficacy was evaluated in a mouse model of myocardial infarction [98]. Compared with control exosomes, the bioengineered exosomes exhibited greater protection regarding the enhancement of angiogenesis in infarcted hearts. This effect may be mediated by an increase in endothelial cell migration and capillary formation through the increase in NOX2-derived ROS. Bone marrow MSCs derived exosomes were electroporated with miR-132 mimics and tested for their ability to promote angiogenesis [99]. In vivo, miR-132-loaded exosomes restored cardiac function and promoted angiogenesis better than did control exosomes following their intramyocardial injection. Kang et al. reported that peripheral blood-derived exosomes could be used for the delivery of miR21 mimic or inhibitor and change the expression of SMAD7, PTEN, and MMP2 in cardiomyocytes. In vivo, miR-21 inhibitor-loaded exosomes could also reduce cardiac fibrosis in a mouse model of myocardial infarction [100]. A distinct advantage of using this approach to deliver therapeutic cargo is the ability to precisely control the amount of drug carried by the EVs. The delivery of miRNA to EVs through electroporation is relatively efficient.

Several EVs-based heart disease therapies have exhibited good performance in experimental models. However, the high risk of off-target effects is a major barrier to the introduction of heart disease therapies in clinics. Thus, several studies have attempted to increase the cardiac targeting of EVs. Exosomes loaded with siClathrin via electroporation significantly reduced the uptake of exosomes by macrophages in vitro and in vivo and increased the exosomes’ delivery to the heart [101]. Preinjection of exosomes containing siClathrin considerably improved therapeutic effects following by injection of miR-21a encapsulated exosomes in a Dox-induced mouse model. The delivery of therapeutic molecules by enhancing targeted EVs provides more possibilities for further clinical applications. The primary advantage of this strategy is that an invasive injection to the heart can be avoided if intravenous administration is employed. Liu et al. employed a nanoparticle “vesicle shuttle” for accurate capture and targeted delivery of circulating exosomes [102]. The nanoparticles consisted of a Fe3O4 core and a silica shell that conjugated through hydrazone bonds to CD63 antigens; CD63-expressing exosomes were captured and accumulated in infarcted cardiac tissue under a local magnetic field. Improvements in infarct size and angiogenesis were observed after nanoparticle injection in both rabbit and rat models. However, it is not easy to achieve targeted delivery of EVs by surface modification, and it requires strict control of the reaction conditions to avoid the destruction and aggregation of EVs due to improper temperature, pH, and osmotic pressure.

Despite less research, the EVs membrane has been chemically modified to enhance the targeting ability of engineered EVs with peptides or proteins. Although the mechanism of interaction with the myocardium is unclear, CHP has exhibited cardiac targeting [103]. Researchers have thus developed a method of conjugating exosomes and CHP through the DOPE-NHS linker and tested the repair effects on myocardial infarction in an ischemia-reperfusion rat model [104]; the engineered exosomes were discovered to increase retention in heart sections, reduce fibrosis and scar formation, and promote cellular proliferation and angiogenesis. Dibenzyl cyclooctyne groups were introduced to the surface of hypoxia-elicited MSCs-derived exosomes (Hypo-Exo) and then interacted with azide-functionalized ischemic myocardium-targeted (IMT) peptide to form stable triazide linkages [105]. The IMT-conjugated Hypo-Exo exhibited specific targeting of the ischemic myocardium due to its interaction with cTnI, which is specifically expressed in ischemic cardiomyocyte. A considerable improvement in heart function and reduced apoptosis and fibrosis were observed in a mouse model of myocardial infarction. In another study, Antes et al. designed a more general EVs membrane anchoring platform based on streptavidin by using DMPE-PEG exosome modification platform; this achieved coupling of any biotinylated molecule to modify the EVs membrane for targeted delivery [106]. CDCs-EVs were then decorated with the ischemia homing peptide (CSTSMLKAC) by using the platform. The CDCs-EVs demonstrated considerably higher localization in injured myocardium when intravenously injected. It should be noted that chemical modification of EVs by using click chemistry may result in nonspecific site reaction due to uncontrolled amino or protein modification. Therefore, there is a risk that chemical modification may shield some protein interactions on the surface of EVs.

The combination of hydrogels and EVs provides a novel approach for optimizing the therapeutic effect of functional EVs. Hydrogels are biocompatible and enable controlled release of encapsulated cargo. Bone marrow MSCs-derived EVs were incorporated into alginate hydrogel (EVs-Gel) and applied in rat models of myocardial infarction. Compared with the control group, the EVs-Gel treatment group markedly decreased cardiomyocyte apoptosis and boosted the polarization of macrophages, which improved cardiac function and infarct size [107]. Similarly, dendritic-cell-derived exosomes could activate Treg cells and promote the polarization of macrophage when they were incorporated into alginate hydrogel, thereby promoting the recovery of cardiac function after myocardial infarction [108]. Additionally, Chen et al. designed a shear-thinning gel (STG) based on the interactions between adamantane and β-cyclodextrin-modified hyaluronic acid [109]. The STG enabled the slow elution of endothelial-progenitor-cells-derived EVs in the ischemic border zone, resulting in higher proliferation of peri-infarct angiogenesis. Further research has indicated that when STG encapsulating EVs were injected on the fourth day after myocardial infarction, they improved left ventricular contractility [110]. Therefore, the therapeutic effects of STG encapsulating EVs can be optimized through strategic timing. Overall, hydrogels can increase the retention of EVs in the heart and enable their controlled release, thus improving their therapeutic effects.

### 4.3. Two-Step EVs Modification Strategy

In the two-step EVs delivery method, the parent cell is genetically modified, and the therapeutic cargo are then loaded into EVs. This composite modification strategy provides more possibilities for targeted delivery and enhancing the efficacy of EVs. Although in infancy, some researchers have conducted exploratory studies on this approach (Figure 3).

Kim and colleagues designed HEK-293 cells expressing cardiac-targeting peptide (CTP) fused to LAMP2b and then isolated EVs from the cell culture medium [111]. The EVs were loaded with an siRNA that targeted the receptor for advanced glycation end products (siRAGE). The modified EVs exhibited twice the siRAGE delivery efficiency of normal EVs and significantly attenuated cardiac inflammation levels in experimental autoimmune myocarditis. In another study, CD47-modified EVs were designed to solve the rapid clearance of EVs by the mononuclear phagocyte system [112]. Binding of CD47 and SIRPα results in a “do not eat me” signal, which helps cancer cells to evade phagocytosis [113]. EVs were generated from MSCs overexpressing CD47 and then loaded with miR-21a via electroporation. After being injected into an ischemia–reperfusion mouse model, the engineered EVs preferentially accumulated in the heart and significantly reduced the apoptosis and inflammation levels of injured cardiomyocyte. It is an attractive strategy to prevent immune rejection of EVs through modulating EVs’ imprinting. Shao et al. isolated exosomes from β2-microglobulin (B2M)-deficient MSCs and loaded miR-24 into exosomes through electroporation [114]. The modified exosomes were able to restore cardiac function by inhibiting cardiac fibrosis in a rat model of myocardial infarction.

### 4.4. Biomimetic EVs for Cardiac Applications

Low yields and complicated purification processes are the main drawbacks to the clinical translation of EVs. In addition to directly or indirectly modifying EVs, scholars have applied biomimetic vesicles in cardiovascular diseases and this approach has recently garnered attention. Biomimetic EVs are synthetic particles that are engineered using the basic moieties present in natural EVs (membrane structures, proteins, lipids, and RNA) to reproduce the therapeutic effects of EVs. This approach facilitates standardization and can improve the yield of EVs because there are a wide range of sources available for the EVs production. Studies have obtained strikingly promising results with biomimetic EVs in cardiac repair processes.

Wang et al. employed a liposome extruder to generate extruded nanovesicles (NVs) from MSCs and tested the repair function of these NVs in a myocardial ischemia–reperfusion mouse model [115]. Similar to natural EVs, extruded vesicles promoted angiogenesis and cardiomyocyte proliferation in the heart after injury and reduced scarring but with the advantage that their yield is more than 20 times that of natural EVs. In a study, adipose MSCs-derived biomimetic NVs were generated by loading melatonin into MSC-extruded NVs through ultrasonic treatment [116]. The engineered NVs significantly restored cardiac function by alleviating mitochondrial dysfunction, reducing fibrosis, and promoting microvessel formation.

Another promising direction regarding biomimetic vesicles is expanding their targeting ability by using membrane-based engineering methods. In one study, mimetic extracellular NVs were extruded from iron oxide nanoparticles-incorporated MSCs [117]. Under magnetic guidance, the NVs exhibited increased retention in an infarcted heart, which consequently reduced apoptosis and fibrosis and improved angiogenesis and cardiac function recovery. Membrane fusion is a convenient method for giving EVs new properties. Li et al. designed platelet-mimetic EVs (P-EVs) by fusing the membranes of MSCs-derived EVs with platelet membranes through extrusion; their therapeutic effect was tested in a mouse model of myocardial ischemia–reperfusion [118]. The P-EVs exhibited significant myocardial targeting and favorably promoted angiogenesis. Furthermore, the P-EVs facilitated the phenotype transformation of macrophages M1 to M2, which may have been due to the release of functional miRNAs [119]. To utilize the recruitment feature of monocytes after myocardium ischemia-reperfusion, Zhang et al. modified MSCs-derived EVs with monocyte mimics through membrane fusion. The engineered EVs exhibited significantly enhanced myocardial targeting capabilities that promoted endothelial maturation for angiogenesis and modulated macrophage subpopulations [120]. These methods are noninvasive and thus clinically appealing.

Biomimetic strategies can be used to increase the retention of EVs in the blood. Avoiding the elimination of EVs by the mononuclear phagocyte system has been a focus of research. Researchers used self-assembly of MSCs membrane on the surface of miR-21-loaded mesoporous silica nanoparticle to construct an exosome-mimicking nanocomplex [121]. This nanocomplex could escape the clearance of the immunologic system and target injured myocardium. Furthermore, the nanocomplex delivered miR-21 to cardiomyocytes, which inhibited the translation of the PDCD4 and PTEN to prevent apoptosis. Platelet membrane-cloaked nanoparticles also exhibited reduced phagocytosis by the mononuclear phagocyte system [122]. Given the biological characteristics of platelet membrane, Li et al. prepared biomimetic Treg nanoparticles by camouflaging cyclosporine A and poly (5,5-dimethyl-4,6-dithio-propylene glycol azelate) within the platelet membrane through extrusion [123]. The biomimetic nanoparticles actively accumulated in the ischemic myocardium, increased the ratio of Treg and M2 type macrophages, and reduced ROS production in a mouse model of myocardial ischemia–reperfusion. The remodeling of the left ventricle and heart function were significantly improved.

## 5. Discussion

Various source-derived EVs have been used for myocardial repair, and each type of EVs has its own advantages. Stem-cells-derived EVs are abundant and rich in therapeutic cargo. Platelet membrane has a natural ability to target an injured vascular wall after myocardial injury. Monocyte-derived membrane is less extensively phagocytized by the mononuclear phagocyte system than are other EVs. Currently, two methods are available for administrating EVs: intramyocardial injection, which is efficient but invasive, and intravenous injection, which is noninvasive but results in low cardiac retention. Further research is necessary to develop noninvasive and efficient EV delivery for clinical applications. The delivery of therapeutic agents by bioengineered EVs leads to greater stability, a longer blood circulation time, and greater agent accumulation in the injured myocardium, all of which enhance the efficacy of the cargo and reduce the dosage required. Compared with the single-step modification strategy, the two-step EVs delivery method has garnered much attention. It works by genetically modifying the parent cell followed by the loading of the therapeutic cargos. The delivery of functional molecules can be optimized, leading to better targeting and lower dosage requirements.

EVs are rich in content, and the various molecules within them have been shown to contribute to EVs’ efficacy in terms of cardiac function recovery. Among their bioactive components, miRNA is the cargo that has most widely been identified as exerting therapeutic effects. However, exogenous nucleic acid insertion, especially that of large messenger RNA, is inefficient. Therefore, an EVs-based nucleic-acid carrier platform should be developed. Additionally, other noncoding RNAs, such as lncRNAs and circRNAs, play functional roles in cardiac recovery. Studies have highlighted the role of specific single therapeutic cargo in EVs; therefore, we recommend further research on the synergistic therapeutic effects of multiple molecules in EVs, such as miRNA clusters.

A variety of mechanisms are involved in the repair process of EVs after myocardial injury. In myocardial infarction, cardiomyocyte apoptosis is the most common event. EVs have a direct inhibitory effect on cardiomyocyte apoptosis. Beyond that, the inhibition of ferroptosis, senescence, autophagy, and pyroptosis is also involved in the cardioprotective effect of EVs [124]. Moreover, regulation of mitochondrial function contributes to EVs-mediated repair. In addition to directly inhibiting apoptosis, EVs promote the recovery of cardiac function by modulating the phenotype of macrophages, a function that can be attributed to the unique miRNAs of EVs [50]. In endothelial cells, EVs promote the formation of new blood vessels. Additionally, fibroblasts can lead to antifibrosis by priming with EVs [125]. Overall, EVs-mediated repair after myocardial injury is a complex process and often combines multiple pathways (Figure 4).

To overcome the limitations of natural EVs, various engineering approaches have been developed. Currently, the engineering modification strategies for EVs have a few primary goals. First, several studies are performed to increase the content of therapeutic molecules in EVs through genetic or nongenetic approaches. It is a useful strategy to identify the potential key EVs miRNAs and their downstream target gene clusters via miRNA chips and computational approaches. Then, the therapeutic cargos can be loaded into EVs through modification of parent cells or isolated EVs. Second, increasing the cardiac targeting of EVs and reducing their clearance by the body’s defense system can enhance the recovery function of EVs. EVs targeting is mainly increased by modifying the surface of EVs with heart-targeting homing peptides or ligands. The combination of hydrogel and EVs leads to localization of the EVs at the border of myocardial injury. Additionally, large-scale production of EVs is essential for clinical translation. A three-dimensional cell culture system is a potential candidate. A microfiber culture platform was found to enrich MSCs-derived particles 1009-fold compared with conventional two-dimensional culture but to also preserve their proangiogenic properties [126]. The biomimetic strategy-extrusion or membrane fusion-may be another means of increasing the EV yield. Notably, biomimetics may improve the standardization, scalability, automatization, cost-efficiency, biological efficacy, and safety of EVs-based cardiac therapy [127]. Overall, the application of these modification strategies can amplify the regenerative effects of EVs.

## 6. Conclusions

In conclusion, EVs therapy is an attractive cell-free approach for treating heart disease. The EVs-based drug delivery platform is a desired pharmacokinetics system. Attempts to modify EVs to improve their biodistribution and cardiotropic properties are still in their infancy. Further studies on EVs biogenesis, content, transfer, and interactions are necessary to develop engineering strategies for specific needs.

## Figures and Tables

**Figure 1 pharmaceutics-14-01848-f001:**
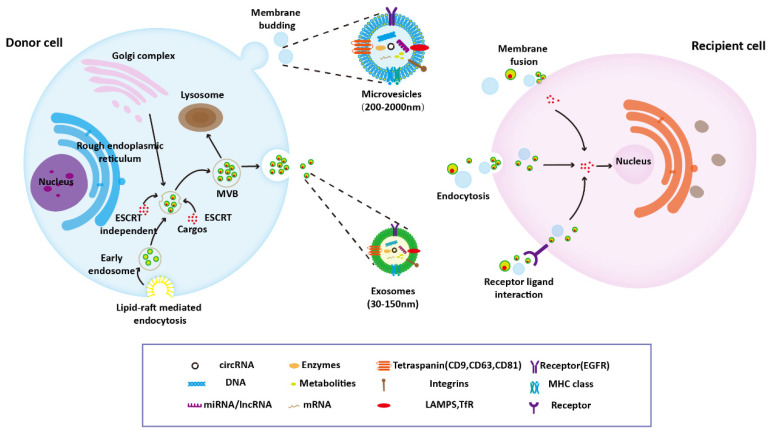
Classification, biogenesis, release, and uptake of EVs. Several stages–including endocytosis, endosome and MVBs formation, and exosomes release–are involved in the formation of exosomes. Microvesicles are released into the extracellular space through outward sprouting of the plasma membrane. EVs deliver their contents to recipient cells through endocytosis, ligand–receptor interaction, and direct fusion with the plasma membrane.

**Figure 2 pharmaceutics-14-01848-f002:**
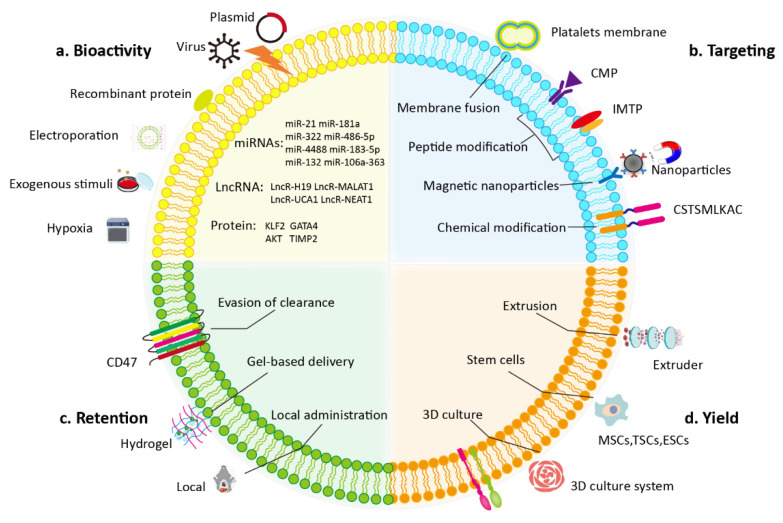
Modulation of EVs for cardiac therapies. Various engineered approaches have been developed to amplify the therapeutic effects of EVs by increasing the bioactivity, targeting, retention, and yield of EVs. Manipulating parental cells or direct modification of the isolated EVs is involved in bioengineering of EVs. Both genetic and nongenetic methods participate in modification strategies.

**Figure 3 pharmaceutics-14-01848-f003:**
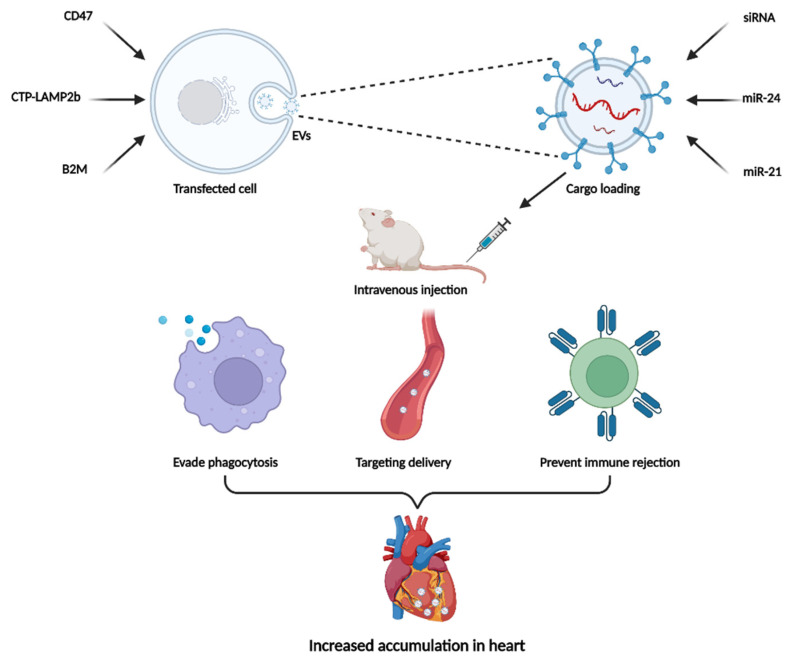
Two-step EVs delivery method for cardiac therapies. Two-step EVs delivery method works by genetically modifying the parent cell followed by the loading of the therapeutic cargos. CTP-Lamp2b, CD47, and B2M-modified EVs were loaded with siRAGE, miR-21, and miR-24 to impart the property of targeting delivery, evading phagocytosis and preventing immune rejection, respectively.

**Figure 4 pharmaceutics-14-01848-f004:**
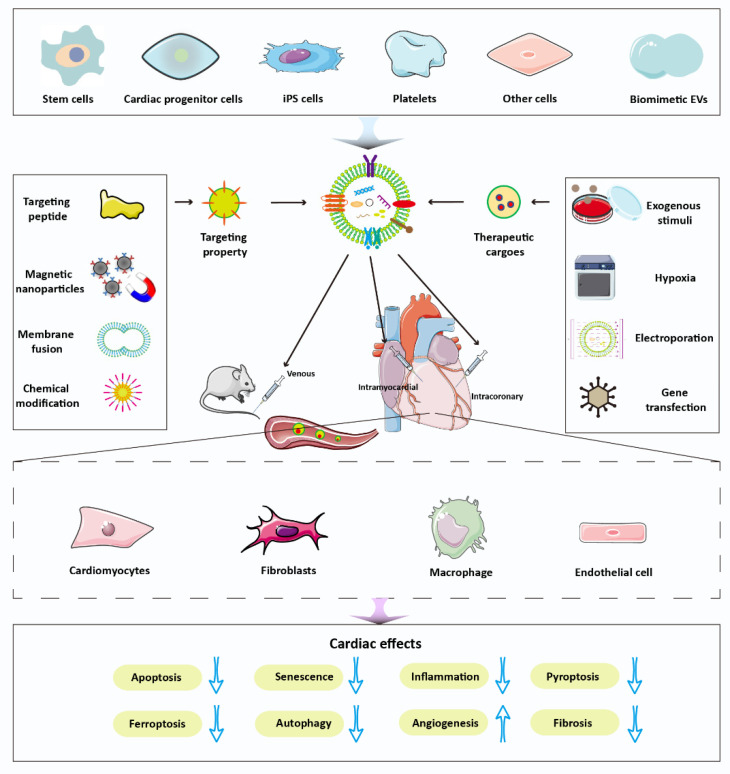
Examples of EVs-modulation strategies for cardiac therapies. Different sources derived-EVs can be modified to increase the therapeutic cargoes and targeting ability for cardiac repair through various injection ways. Then, EVs are able to regulate multiple effect cells including cardiomyocyte, fibroblast, macrophage, and endothelial cells for cardioprotective effects through multiple pathways.

**Table 1 pharmaceutics-14-01848-t001:** Role of natural EVs from different sources for cardiac therapies.

Source	Cargo	Model	Injection	Therapeutic Effects and Mechanism	Reference
BMSCs	miRNAs	MI	Intramyocardial	Inhibit cardiac fibrosis and inflammation and improve cardiac function	[46]
UCB-MSCs	miR-23a-3p	AMI	Intramyocardial	Inhibit ferroptosis and reduce myocardial injury through suppressing DMT1 expression	[47]
UC-MSCs	PINK1	Sepsis	Intraperitoneal	Restore cardiac function and mCa2^+^ efflux-induced mitochondrial calcium overload by PINK1-PKA-NCLX axis	[48]
BMSCs	BMP2	AMI	Intramyocardial	Promote angiogenesis, reduce myocardial fibrosis and inflammatory cell infiltration	[49]
BMSCs	miR-182	I/R	Intramyocardial	Reduce infarct size and alleviate inflammation through modifying the polarization of macrophages M1 to M2	[50]
BMSCs	miR-143-3p	Cellular H/R		Inhibit H/R-induced cell apoptosis and cell autophagy via CHK2-Beclin2 pathway	[51]
MSCs	miR-199a-3p	DOX-induced cardiomyopathy	Intravenous	Inhibit cardiomyocyte apoptosis by the regulation of Akt activation	[52]
BMSCs	miR-29a-3p	Severe acute pancreatitis	Intravenous	Reduce inflammatory markers and improve cardiac function	[53]
UC-MSCs	miR-100-5p	DOX-induced cardiomyopathy		Inhibit DOX-induced ROS, oxidative stress, and apoptosis through the miR-100-5p/NOX4 pathway	[54]
UC-MSCs	lncRNA UCA1	I/R	Intravenous	Inhibit cardiomyocyte apoptosis and promote angiogenesis through the miR-143/Bcl-2/Beclin-1 axis	[55]
BMSCs	LncRNA HCP5	I/R	Intramyocardial	Decrease apoptosis of cardiomyocytes via sponging miR-497 to disinhibit IGF1/PI3K/AKT signaling	[56]
UC-MSCs	circRNAs	Cellular H/R		Inhibit cardiomyocyte apoptosis through the interactions of circRNA-miRNA-PIK3CD	[57]
iPSC-Pg	miRNAs	MI	Intramyocardial	Increase cell survival, proliferation, and migration through specific miRNA signature and cardioprotective pathways	[58]
TSCs		DOX-induced cardiomyopathy	Intravenous	Reduce apoptosis and inflammation with the increase of ZEB1 expression and inhibition of miR-200b	[59]
TSCs	miRNA let-7i	DOX-induced cardiomyopathy	Intramyocardial	Inhibit myocardial apoptosis and decrease inflammatory responses via the let-7i/YAP pathway	[60]
ESCs	miR-294	AMI	Intramyocardial	Promote cardiac neovascularization and reduce apoptosis and fibrosis via the delivery of miR-294 to cardiac progenitor cells	[61]
RAC	miRNAs	I/R	Intravenous	Enhance cardiac function through delivery of key angiogenesis, antifibrosis, anti-inflammatory miRNAs	[63]
CPCs	miR-146a-5p	DOX/Trz-induced cardiomyopathy	Intravenous	Prevent ROS production, reduce inflammatory cell infiltration and fibrosis by suppressing target genes Traf6 and Mpo	[65]
CPCs	miR-210/miR-132	MI	Intramyocardial	Inhibit cardiomyocytes apoptosis and enhance angiogenesis	[66]
CDCs		MI	Intramyocardial	Reduce scar formation, prevent adverse remodeling, and improve left ventricular ejection fraction	[67]
iPS		MI	Intramyocardial	Improve left ventricular ejection fraction, cardiac hypertrophy, cell apoptosis, and angiogenesis	[68]
Plasma	HSP70	I/R	Intravenous	Decrease infarct size and inhibit myocardial apoptosis through the activation of HSP70/TLR4 communication axis	[71]
Plasma	miR-486	I/R	Intracoronary	Protect cardiomyocytes from apoptosis by mediating the inhibition of PTEN, which then activates AKT	[72]
Plasma	miRNAs	MI	Intramyocardial	Promote angiogenesis and ameliorate myocardial remodeling by activating AKT signaling	[73]
MSCs		MI	Spray	Reduce fibrosis and promote cardiac angiomyogenesis	[74]
iPS	miRNAs	AMI	Hydrogel patch	Promote the recovery of ejection fraction, decrease arrhythmia burden, and reduce cardiomyocyte apoptosis	[75]

Abbreviations: BMSCs: Bone marrow MSCs; UCB-MSCs: human umbilical cord blood MSCs; UC-MSCs: umbilical cord MSCs; iPSC-Pg: human induced pluripotent stem-cell-derived cardiovascular progenitors; TSCs: human trophoblast stem cells; ESCs: embryonic stem cells; RAC: regeneration-associated cells; CPCs: cardiac-resident mesenchymal progenitor cells; CDCs: cardiosphere-derived cells; iPS: induced pluripotent stem cells.

## Data Availability

Not applicable.

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
