# Peer review of "Extracellular Vesicles: A New Frontier for Cardiac Repair"

_pharmaceutics, 2022, doi:10.3390/pharmaceutics14091848_

Round 1
Reviewer 1 Report
In this interesting review, You and colleagues focus their attention on the impact of EV as new therapeutic tools in cardiac repair. The manuscript is well written and the figures are really evocative. A point that should be fixed is related to the size of different EV: exosomes are 30-150 nm. The field of research focused on exosomes is in continuous evolution, and even if the related section is well written, the references could be updated with more recent works related to the need for new technologies for the association of a specific marker with an exosome subtype and the exosome subtype to a particular function and/or group of functions ( PMID: 35141731 and others). The overall consideration goes in the direction of a really good paper and I feel that also taking into consideration the above comments the paper could be appreciated by the Pharmaceutics readers. Good luck.
Author Response
Dear Editor,
Thank you for your e-mail dated Aug 25, 2022 regarding the review of our manuscript. We appreciate your assessments of the manuscript and have found that the comments and suggestions are helpful in preparation of the revised manuscript. We revised the manuscript as suggested by reviewers. In the revised manuscript, all the amendments were highlighted by blue words. The following are our point-by-point responses, in order of the comments about the manuscript:
Q: The manuscript is well written and the figures are really evocative. A point that should be fixed is related to the size of different EV: exosomes are 30-150 nm.
Responses: Thanks for your positive comments and suggestion. We have corrected it in our revised manuscript.
Q: The field of research focused on exosomes is in continuous evolution, and even if the related section is well written, the references could be updated with more recent works related to the need for new technologies for the association of a specific marker with an exosome subtype and the exosome subtype to a particular function and/or group of functions (PMID: 35141731 and others).
Responses: Thanks for your kind suggestion. We have updated some references in the revised manuscript to accurately reflect the latest research progress. The reference you suggested gives us a good example.
We hope that the above responses meet the expectations of the reviewer. The comments and suggestions were helpful in improving the quality of our manuscript.
Thank you for your consideration. We look forward to hearing from you and to publication of this manuscript.
Sincerely,

Reviewer 2 Report
This manuscript aims to highlight the growing field of EVs as therapeutic agents for cardiac repair. It discusses the uses for both native and engineered EVs as therapeutic delivery modalities and discusses the advantages and disadvantages with each approach. Overall, the topic is very relevant, and the manuscript is interesting. The discussion in particular discusses relevant considerations with respect to EV administration and consideration of EV use as therapeutics. This is a strength of the manuscript. Additional thoughts are listed below:
· The discussion on nomenclature in the biosynthesis and characteristic of EVs section is very informative for novices in this area. However, it should go a bit further to explain some concepts. For example, the term exosome should only be used when data demonstrating endosomal origin of an EV is provided. It would also be helpful to explain the relationship between exosomes, microvesicles, and EVs. Finally, discuss how the terminologies are used in the current manuscript.
· The International Society of Extracellular Vesicles is working to help standardize the young EV field. It would likely be helpful to cite some of their efforts toward a uniform consensus, particularly in relation to nomenclature.
MISEV 2014: https://www.tandfonline.com/doi/full/10.3402/jev.v3.26913
MISEV 2018: https://onlinelibrary.wiley.com/doi/10.1002/jev2.12182
· Line 71-2: Supermeres (and similarly exomeres) are non-vesicular extracellular nanoparticles.
· Table 1 is a bit confusing. It is helpful to tabulate the role of natural EV in cardiac therapies but at the same time the table is not a comprehensive list of the role of EVs for cardiac therapeutics (as this would be an extensive list). It would be helpful to be clearer about the point being made here.
· Gao et al. also demonstrated the utility of hiPSCs EVs in recovery from myocardial infarction. https://doi.org/10.1126/scitranslmed.aay1318. Barile et al. https://doi.org/10.1093/cvr/cvu167
· Formatting could also be improved. Examples include lines 32, 37, 502, figures (fig1), and text reference spacing among others.
Author Response
Dear Editor,
Thank you for your e-mail dated Aug 25, 2022 regarding the review of our manuscript. We appreciate your assessments of the manuscript and have found that the comments and suggestions are helpful in preparation of the revised manuscript. We revised the manuscript as suggested by reviewers. In the revised manuscript, all the amendments were highlighted by blue words. The following are our point-by-point responses, in order of the comments about the manuscript:
Q: The discussion on nomenclature in the biosynthesis and characteristic of EVs section is very informative for novices in this area. However, it should go a bit further to explain some concepts. For example, the term exosome should only be used when data demonstrating endosomal origin of an EV is provided. It would also be helpful to explain the relationship between exosomes, microvesicles, and EVs. Finally, discuss how the terminologies are used in the current manuscript.
Responses: Thanks for your suggestion. These concepts on EVs are useful to readers, especially non-specialists on the subject. In the revised manuscript, we have more clearly stated the relevant concepts and made necessary explanations for the terminologies used in the article.
Q: The International Society of Extracellular Vesicles is working to help standardize the young EV field. It would likely be helpful to cite some of their efforts toward a uniform consensus, particularly in relation to nomenclature.
Responses: Thanks for your suggestion. We have cited relevant reference in our revised manuscript.
Q: Line 71-2: Supermeres (and similarly exomeres) are non-vesicular extracellular nanoparticles
Responses: Thank you for your correction. We have corrected this statement.
Q: Table 1 is a bit confusing. It is helpful to tabulate the role of natural EV in cardiac therapies but at the same time the table is not a comprehensive list of the role of EVs for cardiac therapeutics (as this would be an extensive list). It would be helpful to be clearer about the point being made here.
Responses: We are sorry for causing this confusion. Table 1 is a summary of the functional roles of various sources-derived natural EVs mentioned in the manuscript on cardiac repair. In addition to the emphasis on different EVs sources, content that is lacking in the main text is supplemented, such as the injection methods of EVs. We hope to provide readers a more concise and comprehensive understanding of the effect of natural EVs on cardiac therapy in the form of table.
Q: Gao et al. also demonstrated the utility of hiPSCs EVs in recovery from myocardial infarction. https://doi.org/10.1126/scitranslmed.aay1318. Barile et al. https://doi.org/10.1093/cvr/cvu167.
Responses: Thanks for your suggestion. We have cited these references in our revised manuscript.
Q: Formatting could also be improved. Examples include lines 32, 37, 502, figures (fig1), and text reference spacing among others.
Responses: Thank you for your reminder. We have corrected these problems in the revised manuscript.
We hope that the above responses meet the expectations of the reviewer. The comments and suggestions were helpful in improving the quality of our manuscript.
Thank you for your consideration. We look forward to hearing from you and to publication of this manuscript.
Sincerely,
